# Cyclic bouts of extreme bradycardia counteract the high metabolism of frugivorous bats

M Teague O'Mara[1,2,3,4]*, Martin Wikelski[1,2], Christian C Voigt[5], Andries Ter Maat[6], Henry S Pollock[7], Gary Burness[8], Lanna M Desantis[9], Dina KN Dechmann[1,2,3]

[1]Department of Migration and Immuno-ecology, Max Planck Institute for Ornithology, Radolfzell, Germany; [2]Department of Biology, University of Konstanz, Konstanz, Germany; [3]Smithsonian Tropical Research Institute, Panama City, Panama; [4]Zukunftskolleg, University of Konstanz, Konstanz, Germany; [5]Leibniz Institute for Zoo and Wildlife Research, Berlin, Germany; [6]Department of Behavioural Neurobiology, Max Planck Institute for Ornithology, Starnberg, Germany; [7]Program in Ecology, Evolution and Conservation Biology, University of Illinois at Urbana-Champaign, Urbana, United States; [8]Department of Biology, Trent University, Peterborough, Canada; [9]Environmental and Life Sciences Graduate Program, Trent University, Peterborough, Canada

**Abstract** Active flight requires the ability to efficiently fuel bursts of costly locomotion while maximizing energy conservation during non-flying times. We took a multi-faceted approach to estimate how fruit-eating bats (*Uroderma bilobatum*) manage a high-energy lifestyle fueled primarily by fig juice. Miniaturized heart rate telemetry shows that they use a novel, cyclic, bradycardic state that reduces daily energetic expenditure by 10% and counteracts heart rates as high as 900 bpm during flight. *Uroderma bilobatum* support flight with some of the fastest metabolic incorporation rates and dynamic circulating cortisol in vertebrates. These bats will exchange fat reserves within 24 hr, meaning that they must survive on the food of the day and are at daily risk of starvation. Energetic flexibly in *U. bilobatum* highlights the fundamental role of ecological pressures on integrative energetic networks and the still poorly understood energetic strategies of animals in the tropics.
DOI: https://doi.org/10.7554/eLife.26686.001

*For correspondence:
teague.omara@gmail.com

Competing interests: The authors declare that no competing interests exist.

## Introduction

Energy intake, incorporation and expenditure are fundamental to animal behavior and evolution (*Brown et al., 2004*; *Weiner, 1992*). Animals must balance between generating enough metabolic power to find and acquire food and maintaining sufficient reserves to sustain daily maintenance, and repair and reproduce. This basic requirement of life can drive the foraging strategies of entire clades (*Williams et al., 2014*) and extensive links among various behavioral and physiological strategies have evolved in response to single ecological pressures including diet and pathogen environments (*Cohen et al., 2012*). This is largely a consequence of the sequential and linear process of energetic input (feeding), and that energy expenditure is additive across parallel aspects of physiology (*Weiner, 1992*). Energetic networks then link across physiological systems from mitochondrial oxidation to digestion, to respond to changes in resource availability and maintain physiological integrity. Well-adapted energy metabolisms must then both be able to conserve reserves and deliver

**eLife digest** To survive, all animals have to balance how much energy they take in and how much they use. They must find enough food to fuel the chemical processes that keep them alive – known as their metabolism – and store leftover fuel to use when food is not available. Bats, for example, have a fast metabolism and powerful flight muscles, which require a lot of energy. Some bat species, such as the tent-making bats, survive on fruit juice, and their food sources are often far apart and difficult to find. These bats are likely to starve if they go without food for more than 24 hours, and therefore need to conserve energy while they are resting.

To deal with potential food shortages, bats and other animals can enter a low-energy resting state called torpor. In this state, animals lower their body temperature and slow down their heart rate and metabolism so that they need less energy to stay alive. However, many animals that live in tropical regions, including tent-making bats, cannot enter a state of torpor, as it is too hot to sufficiently lower their body temperature. Until now, scientists did not fully understand how these bats control how much energy they use.

Now, O'Mara et al. studied tent-making bats in the wild by attaching small heart rate transmitters to four wild bats, and measured their heartbeats over several days. Since each heartbeat delivers oxygen and fuel to the rest of the body, measuring the bats' heart rate indicates how much energy they are using. The experiments revealed for the first time that tent-making bats periodically lower their heart rates while resting (to around 200 beats per minute). This reduces the amount of energy they use each day by up to 10%, and helps counteract heart rates that can reach 900 beats per minute when the bats are flying.

Overall, these findings show that animals have evolved in various ways to control their use of energy. Future research should use similar technology to continue uncovering how wild animals have adapted to survive in different conditions. This knowledge will help us to understand how life has become so diverse in the tropics and the strategies that animals may use as climates change.

DOI: https://doi.org/10.7554/eLife.26686.002

enormous energetic power outputs in an efficient and effective manner. However, few animal models currently allow us to follow energy from intake to delivery of energetic currency to fuel metabolism, and finally to the countermeasures taken to slow down energetic expenditure when it is not needed. Furthermore, accumulating evidence shows that data collected in laboratory settings may not reflect the full range of strategies animals employ to deal with this energetic dilemma (*Bishop et al., 2015*; *Bowlin et al., 2005*; *Calisi and Bentley, 2009*; *Geiser et al., 2007*; *Ward et al., 2002*). This makes quantitative data from naturally behaving animals in the wild even more important to test the balance and integration of physiological adaptations to energetic limitations.

Flying vertebrates are an excellent example of this balance. While flight is one of the most efficient modes of locomotion per unit distance traveled, it is costlier per unit time than any other mode of locomotion (*Norberg, 1990*; *Schmidt-Nielsen, 1979*). To fulfill the exceptional demands of powered flight, both birds and bats have undergone dramatic physiological reorganization that emphasizes the need to supply fuel to large flight muscles (*Maina, 2000*; *Norberg, 1990*). Bat flight in particular is an extreme case of vertebrate locomotor energetics. In comparison to those of non-flying mammals of comparable size, hearts and lungs of bats are larger and have higher blood oxygen transport potential, delivering more oxygen per heart beat than non-flying terrestrial mammals (*Neuweiler, 2000*). Bats use some of the highest mass-specific metabolic rates during flight; 3–5 times greater than any other mammals and maximum increases of 15–16 times minimum resting metabolic rates (*Speakman and Thomas, 2003*). This may place bats at their energetic ceiling, and integrated physiological networks that allow them to maintain high metabolic rates at or near their limits over extended periods of time may be under equally strong selection to reduce resting energetic expenditure below what is commonly found in mammals.

Bats launch themselves directly into energy-demanding flight at the onset of their activity period and on an empty stomach, fueling flight by limited fat reserves (*Voigt et al., 2010*). They must then efficiently find and ingest food, and make energy available to their metabolism rapidly, as high

metabolic rates and small body size place them at risk of starvation if sufficient food is not found. This risk is enhanced in the many species that specialize on ephemeral food sources. One strategy to cope with this energetic vulnerability is through daily reduction of metabolic rate (torpor) found in small-bodied bat species especially from the temperate zone. By entering a distinct low-energy state characterized by low body temperature, some bats reduce metabolic rates by 99% during torpor when ambient temperatures are lower than their thermoneutral zone (*Geiser and Stawski, 2011*; *Ruf and Geiser, 2015*). In tropical and sub-tropical regions where ambient temperatures are high, it may be impossible to lower body temperature beyond these critical minimum temperatures to save energy, therefore reductions in heart rate may reflect reductions in cellular respiration rates and gene expression in multiple pathways and be an effective measure of energetic conservation (*Dechmann et al., 2011*; *Dzal et al., 2015*; *McNab, 1969*; *Storey and Storey, 2004*). This may be particularly important in those that feed on sugar dense foods as they are at the highest risk of starvation (*McNab, 1969*; *Voigt and Speakman, 2007*).

Heart rate has a quadratic relationship with metabolic oxygen consumption (*Bishop and Spivey, 2013*; *Grubb, 1982*), and by measuring it directly it is possible to gain insight into energetic expenditure at high temporal resolution. Heart rates in bats may more than double in the transition from rest to flight, reflecting enormous flight power requirements (*Thomas, 1975*). Controlled experiments in wind tunnels and laboratory conditions have yielded incredible insight into the regulation of metabolism and energy consumption across a wide variety of activities and physiological states. However, heart rates of exercising animals in nature are unpredictable and metabolic rates measured during wind tunnel flight may not indicate the full scope of natural behavior. In a tropical insectivorous bat, heart rate increases from 129 bpm in the roost to 847 bpm during flight, a six-fold increase that is larger than predicted from other captive bats in wind tunnels (*Dechmann et al., 2011*). Alternatively, heart rates of free-flying animals may be much lower than expected. For example, bar-headed geese traverse the Himalayas with heart rates of 250–475 bpm (*Bishop et al., 2015*), 20% lower than what is expected from captive measures (*Ward et al., 2002*), and during migration, heart rates in Swainson's thrushes are 10% lower than comparable long flights in wind tunnels (*Bowlin et al., 2005*). This indicates that sustainable metabolic rates possible during exercise may differ greatly from maximal rates or extrapolations in captive studies and we have only been able to get an initial glimpse into the heart rates used by flying bats.

Once they begin to feed, bats fuel their enormous demand for power by directly and rapidly metabolizing ingested food, but this can lead to high risk of starvation via rapid fat turnover (*Caviedes-Vidal et al., 2008*; *Voigt and Speakman, 2007*). One mechanism that may help animals to adjust the timing and intensity of shifts in metabolic scopes are glucocorticoids. They are key integrators between the environment and energy balance that ensure rapid response to changes in energetic needs (*Cohen et al., 2012*). Elevated levels of glucocorticoid hormones in blood plasma suppress glycogen formation and promote gluconeogenesis (*Haase et al., 2016*), fat oxidation (*Brillon et al., 1995*), and play a primary role in energy balance (*Nieuwenhuizen and Rutters, 2008*). Most bats that have been studied show high baseline glucocorticoid concentrations (*Reeder et al., 2004*; *2006*), which indicates that they are in a ready state to rapidly mobilize glucose and glycogen reserves. By manipulating circulating levels of glucocorticoids or those tied to binding globulin, individual use of energy reserves can be modulated (*Schneider, 2004*).

Bats are then faced with an energetic dilemma where they must rapidly power flight, but quickly switch to conserving energetic stores gained during foraging. To better understand the interplay of energy expenditure and conservation, we describe the daily energetic life of Peters' tent-making bat (*Uroderma bilobatum*, family Phyllostomidae) in Gamboa, Panamá. These bats are central-place foragers that leave a stable roost location to feed primarily on juice extracted from ripe figs (*Ficus* spp). We hypothesized that they would not use torpor during their regular daily life and that their energy intake and turnover rates would be high. Daily energy intake and expenditure should then be closely matched, resulting in a specialized life-style at the energetic edge. Testing our hypotheses was made possible by newly miniaturized heart rate transmitters to describe both the activity patterns of the species and their instantaneous energetic expenditure throughout the day, including the first flying heart rates of free-ranging individuals. We also tested how these bats fuel their metabolism through measurement of metabolic incorporation rates and fat turnover from stable isotope ratios in their breath in short-term captivity (*Voigt and Speakman, 2007*). In combination with an estimate of

energy mobilization potential via elevated circulating cortisol, this allows us a more complete view into how these small-bodied, high-metabolic frugivores meet daily energetic demands.

## Results

### Activity patterns

We tracked the heart rates (*Figure 1—figure supplement 1*) of four bats for 13.6 ± 4.9 hr (mean ± SD) each day for two to four days (13 days total). This included 4.03 ± 0.05 hr of activity outside of the roost at night and the approximately 12 hr that bats spend in their roost during the day for 350 hr of total recording time. All bats left their roosts between 18:00 – 18:30 and flew three to seven minutes to their initial foraging sites. Bats executed multiple short flights of 1–2 min each (mean ± SD: 1 ± 1.5 min) that were consistent with flying to a fruiting tree, selecting a fruit, and carrying it to a separate feeding perch. During our tracking, flight accounted for 13 ± 6% (30.6 ± 15.6 min) of the time outside of the roost. We were able to locate several food trees, all of which were *Ficus insipida*, but all bats also fed for short periods across the Panama Canal at sites inaccessible during tracking. Bats returned to their day roosts between 22:30 – 06:00. When bats returned early in the night, they left for an additional one to two hours later in the morning. The minimum time that we tracked a bat foraging, including short bouts away from the roosts was two hours and the maximum total time outside of the roost was 10 hr. All bats returned to their home roost each night where they remained for the rest of the day.

### Field metabolic rates and cyclic bradycardia

*Uroderma bilobatum* used a large range in heart rates ($f_H$) across the day, ranging from 173 to 1066 bpm (*Figure 1—figure supplement 2*). Analysis of activity-specific $f_H$ shows that bats expend 4.9 ± 0.8 kJ $h^{-1}$ during flight (mean ± SD; $f_H$: 766 ± 56 bpm, *Figure 1*). Amplitude fluctuations of the $f_H$ radio signals show that minimum $f_H$ of flying bats was 750 bpm. Maximum recorded flying $f_H$ was 1066 bpm. *Uroderma bilobatum* then needed to generate a minimum of 0.98 W to fly (3.5 kJ $h^{-1}$), but flight typically had higher costs of 1.36 ± 0.23 W with a maximum recorded output of 2.3 W. This is a mass-specific metabolic power of 75.89 ± 11.9 W $kg^{-1}$ and a maximum mass specific power of 145.6 W $kg^{-1}$. Nightly non-flight activity when bats were stationary required 2.2 ± 1.1 kJ $h^{-1}$ ($f_H$: 492 ± 128 bpm, *Figure 1*).

Surprisingly, *U. bilobatum* periodically lower $f_H$ to 200–250 bpm from a mean $f_H$ of 374 ± 112 bpm throughout their daily resting periods where they remain relatively inactive in their roosts, and during which they consume 1.2 ± 0.8 kJ $h^{-1}$ (0.33 — 0.23 W or 18.84 ± 13.59 W $kg^{-1}$, *Figure 2*). During these periods bats are typically sitting quietly, although bats can be alert during these times and engage in bouts of agonism, grooming, and may fly from the roost due to disturbances around the roosting sites. Bats suppress $f_H$ by 30% 2–3 times per hour (mean: 1.54 ± 1.18 sd times per hour) for 5–7 min throughout the day (*Figure 2*). This cyclic bradycardia is a yet undescribed strategy that was only detectable through complete sampling of daily heart rate recordings. These lowered heart rates were followed by a return to the more stable rates between 300–400 bpm, or often to a brief arousal state with elevated heart rates above resting rates. All bats employed these reduced heart rates but one individual only used them on two of the four days it was observed.

This lowered heart rate resulted in a median resting metabolic rate (RMR) of 0.54 ± 0.01 kJ $h^{-1}$ compared to RMR 0.75 ± 0.04 kJ $h^{-1}$ at higher mean $f_H$. Using the mean energetic expenditure by each bat on each night it was tracked (*Supplementary file 1*) we can estimate typical field metabolic rate (FMR) of 45.79 kJ if a bat spends 2 hr in flight and executes daily cyclic bradycardia (*Figure 3*). Two hours may be an over-estimate of time flying in the resource dense region where we tracked bats, but likely reflects areas with more dispersed fruit trees. Based on median values for each individual mean metabolic scope was 5.39 ± 1.80. This short, cyclic bradycardia then saved *U. bilobatum* 0.3–0.5 kJ $h^{-1}$ or 3.5–6 kJ total over the 12 hr resting phase which is 10% (7.6–13.1%) of their total FMR.

### Metabolic incorporation rates of resting bats

We used a diet switching experiment that transitioned bats from a natural diet, dominated by figs with low $\delta^{13}C$ values, to an experimental diet with high $\delta^{13}C$ values (agave sugar) to model the

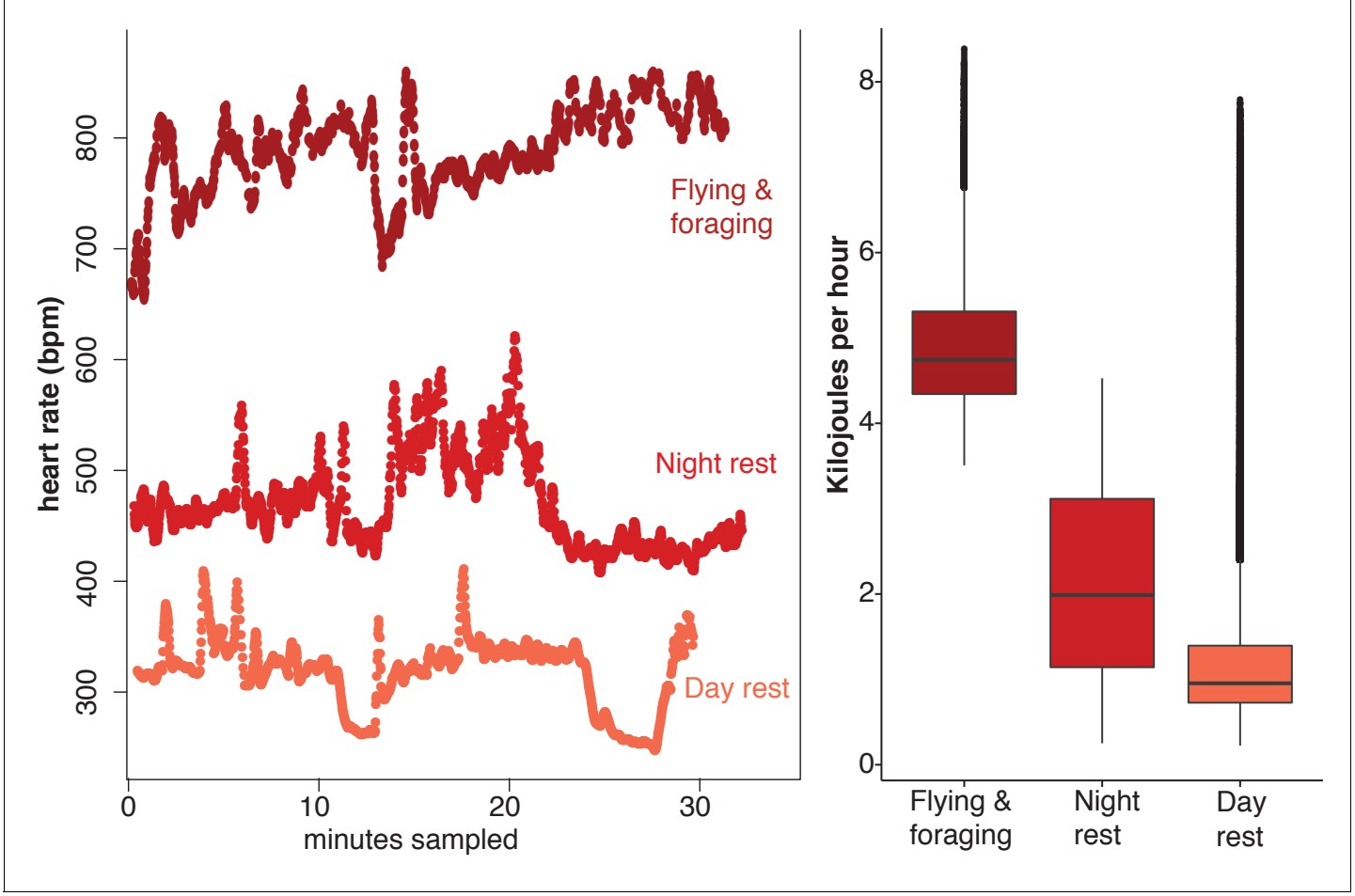

**Figure 1.** Heart rate and energetic expenditure of *U. bilobatum* recorded across 350 hr of observation. (**A**) 30 min examples of continuous heart rates of *Uroderma bilobatum* during daily activities and (**B**) the distribution of energetic costs estimated for these activities from heart rate.

DOI: https://doi.org/10.7554/eLife.26686.003

The following figure supplements are available for figure 1:

**Figure supplement 1.** Processing heart rate ($f_H$) radio signals to extract heart rates of free-ranging bats.

DOI: https://doi.org/10.7554/eLife.26686.004

**Figure supplement 2.** Distribution of heart rates during flying and foraging, nightly non-flight activities, and in-roost rest during day light hours.

DOI: https://doi.org/10.7554/eLife.26686.005

**Figure supplement 3.** Flow through respironmetry calibration of non-exercising *Uroderma bilobatum* heart rate.

DOI: https://doi.org/10.7554/eLife.26686.006

**Figure supplement 4.** Calibration of heart rate versus energy consumption measured in flow through respirometry of non-exercising bats (filled circles and dashed line) versus the energetic expenditure predicted by estimates for exercising animals that incorporate exercise-induced changes in stroke volume (open diamonds, solid line).

DOI: https://doi.org/10.7554/eLife.26686.007

speed at which ingested sugar enters metabolism by measuring the changes in the $\delta^{13}C$ composition of exhaled $CO_2$. After a baseline sample, bats (n = 8) were fed a solution of agave nectar. Their exhaled breath was rapidly enriched in $^{13}C$ and reached an asymptotic value of $-16.5 \pm 2.0$ ‰ 50 min after initial feeding (*Figure 4A*, *Supplementary file 2*) which is lower than the $\delta^{13}C$ value of the diet and indicates that fat or glycogen stores continued to be metabolized ($\delta^{13}C_{diet} = -12.0 \pm 0.1$ ‰, t = −12.6, df = 32, p<0.001). Overall $\delta^{13}C_{breath}$ enrichment followed a mean single pool incorporation model of $\delta^{13}C_{breath}(t) = -16.575 - 12.841e^{-0.081t}$, with 50% of metabolism fueled by ingested food after only 8 min ($t_{50} = 8.1 \pm 15.6$ min). The large standard deviation in $t_{50}$ is due to one distinctive bat (Individual A) that showed a nearly linear enrichment curve with no asymptote (*Supplementary file 2*). If this bat is excluded, $t_{50}$ drops to 7.6 min and an incorporation curve of

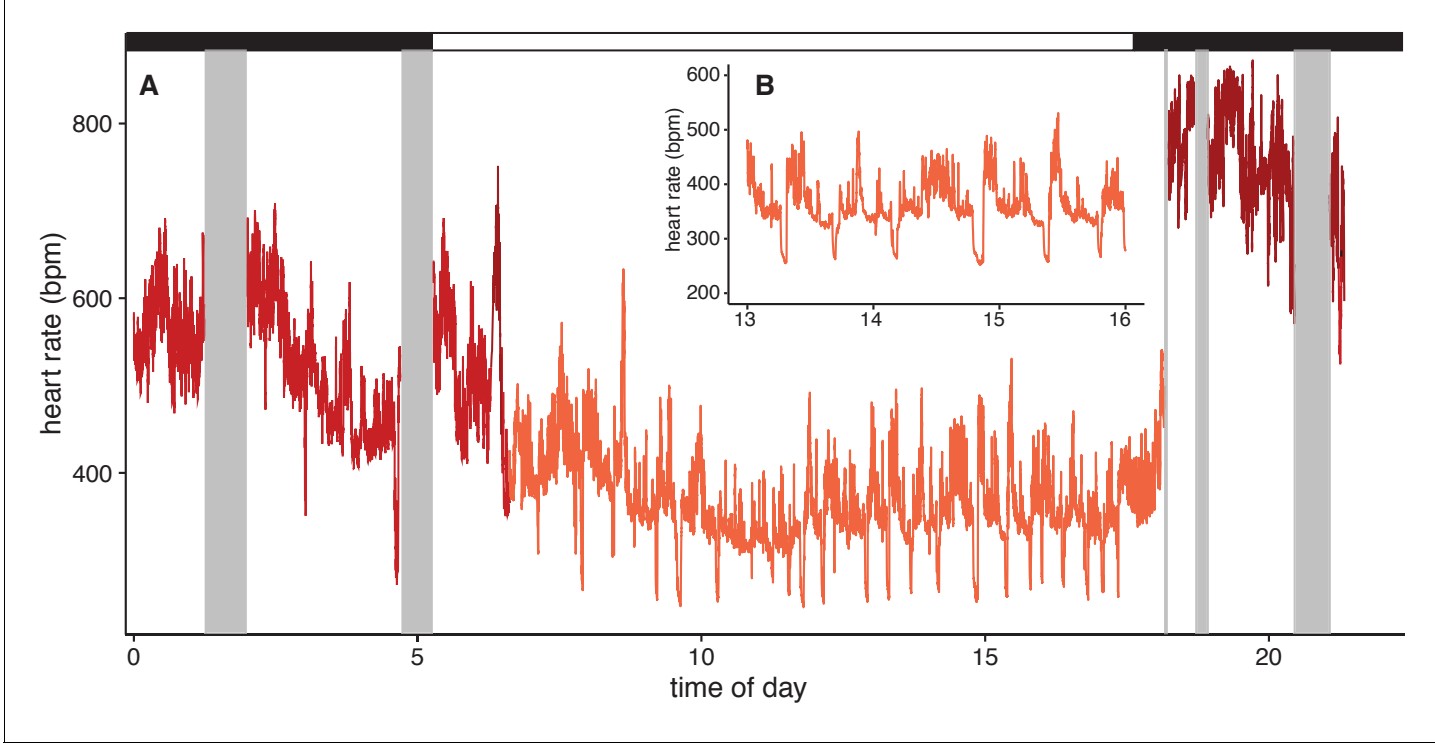

**Figure 2.** Example heart rate recordings of one individual (bat 1) from 2014-12-07. (A) Twenty-four hours of observation include periods of missing data when the bat was out of tracking range (grey boxes). Black and white bars above indicate night and day. Inset B shows more detail from the same time period (13–16 hr) to highlight the daily, cyclic bradycardia executed by these bats that save up to 10% of their daily energetic expenditure.
DOI: https://doi.org/10.7554/eLife.26686.008

$\delta^{13}C_{breath}(t) = -16.497 - 13.138e^{-0.091t}$. Bats fed with ripe *Ficus indica* (n = 6) did not show any change in $\delta^{13}C_{breath}$ over the course of the following 90 min (*Figure 4A*, $F_{1, 24}$ = 2.614, p=0.113).

Over the course of the next three days, bats kept in captivity and fed on agave nectar showed increasingly $^{13}C$ enriched baseline $\delta^{13}C_{breath}$ values at the beginning of the night (*Figure 4B*) and after not eating for the entire day, which is typical of feeding patterns of these bats. We estimated a mean single-pool exponential model of $\delta^{13}C_{breath}(t) = -16.412 - 12.901e^{-0.801t}$, with a $t_{50}$ = 13.2 ± 4.6 hr, and by the third night bats approached an asymptotic starting value of −17.06 ± 1.27 ‰ which is not different from the asymptotic value of the initial feeding experiment ($t_{13}$ = 0.91, p=0.38). This indicates that fifty percent of an individual's fat reserves are then exchanged after 13–17 hr, and a carbon atom has a residency period of 1–2 days, with a full exchange of fat after 3 days.

### Glucocorticoids and energy mobilization

Bats captured at their roosts (15 F, 6 M) showed low baseline values of circulating cortisol concentrations (ng ml$^{-1}$) that did not differ by sex (F: 64.81 ± 158.81 ng ml$^{-1}$, M: 57.66 ± 137.07 ng ml$^{-1}$, $F_{1,19}$ = 0.009, p=0.92). When restrained in a cloth bag for one hour they showed a strongly sex biased response: restraint-induced values were 10–15 times baseline conditions (*Figure 5*), and were two-fold greater in females than in males (F: 989.50 ± 450.78 ng ml$^{-1}$, M: 428.34 ± 94.45 ng ml$^{-1}$, $F_{1,19}$ = 0.8.89, p=0.008).

## Discussion

We tracked the heart rates free-ranging bats throughout the 24 hr period, including foraging, to estimate total energetic costs. As hypothesized, we found that heart rate derived energy expenditure of *U. bilobatum* during flight is high and this is achieved through rapid incorporation of ingested food into their metabolism. We found flying heart rates that were 4–5 times higher than resting rates during the day and twice the heart rates of bats roosting at night. These bats replace nearly half of

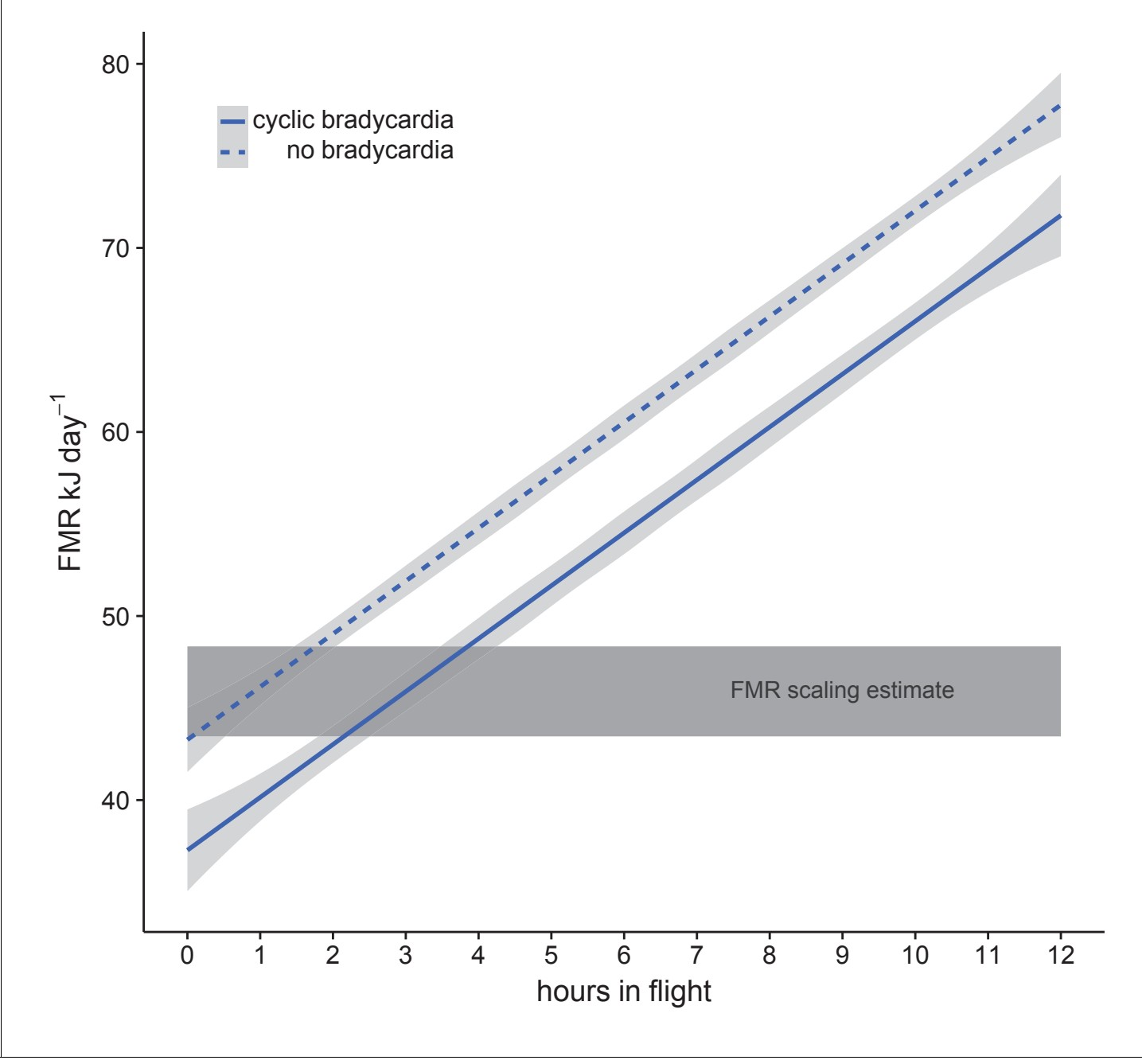

**Figure 3.** Mean field metabolic rate ±95% CI estimated by the number of hours spent in flight with (solid line) and without (dashed line) daily cyclic bradycardia. A conservative estimate of two hours flight and a mean FMR of 45.79 kJ day$^{-1}$ is based on our radio tracking observations of free-flying bats in their natural environment. This is within the estimates from the *Speakman (2005)* scaling relationship (grey box) for the range of body masses (16–19 g) measured in this population.

DOI: https://doi.org/10.7554/eLife.26686.009

their fat reserves within a single day, resulting in short potential starvation times. *Uroderma biloba-tum* counter this high energetic expenditure by spending relatively little time in flight and they have exceptionally low circulating cortisol values at rest during the day. These low basal values promote conservation of glucose reserves, but can be elevated up to 15 times, at least in response to stress, and could be used to generate the high metabolic power needed for flight. Most surprising, we found that by cyclically lowering heart rates during the day, they save 10% of their energy budget.

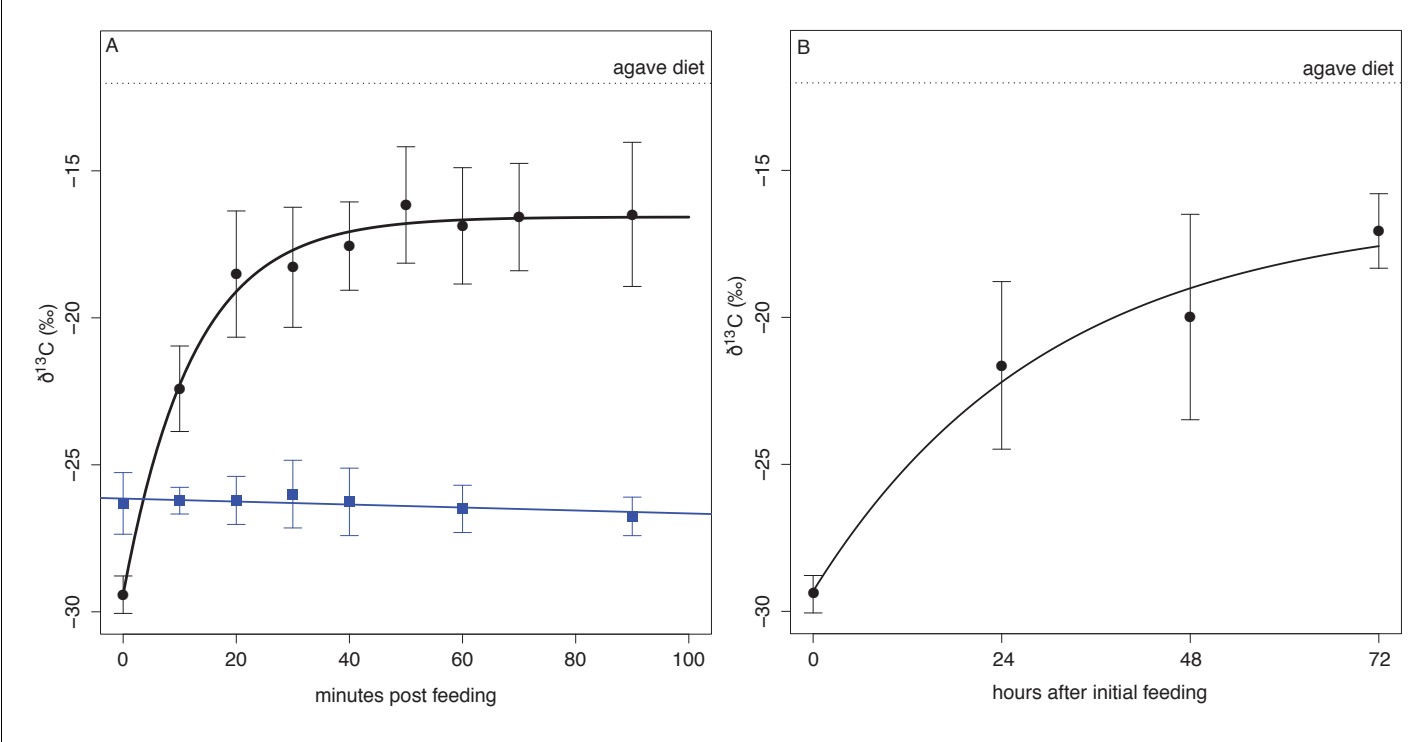

**Figure 4.** $\delta^{13}$C measured from exhaled $CO_2$ post feeding on agave nectar (black circles) and *Ficus insipida* (blue squares). **(A)** *Uroderma* fueled metabolism from ingested food immediately upon feeding on agave nectar (black) and fueled 50% ($t_{50}$) of their metabolism within 8 min. There was no change in $\delta^{13}$C when bats were fed figs that comprise their natural diet. **(B)** When fed agave nectar over 72 hr bats reached a $t_{50}$ for fat replacement after 13 hr and approached asymptotic values at 48 hr.

DOI: https://doi.org/10.7554/eLife.26686.010

This cyclic bradycardia is a novel strategy that minimizes energetic expenditure at relatively high ambient temperatures and allows *U. bilobatum* to maintain a FMR expected for their size. Only by completely sampling these high-resolution data from naturally behaving bats were we able to detect these lowered heart rates and quantify their effect on bat energetic strategies.

Using the energetic expenditure derived from median heart rates of resting bats in their natural roosts during the day (0.54 kJ h$^{-1}$ or 0.27 W) we can estimate a RMR of 13.0 kJ day$^{-1}$, which closely approximates previous measures of BMR (12.8 kJ day$^{-1}$ [*McNab, 1969*]). After commuting to the foraging patch, figs are collected during short flights of 1–2 mins and most of the remaining time is spent more or less at rest in their night feeding roosts resulting in only 30 mins per night in actual flight. Although this may differ among sites or during periods of less favorable food availability, this perch-resting with short fruit collection flights is an important part of their energy saving strategy. The subsequently low heart rates and activity patterns estimate an estimated FMR of ca. 46 kJ day$^{-1}$ (*Bishop and Spivey, 2013*), which is within the general predictions for FMR based on body mass from a broad taxonomic sampling of studies using doubly-labeled water (*Speakman, 2005*). While energetic expenditure met estimates for 16–19 g bats, this was only possible due to *U. bilobatum* restriction of total active flying time to less than about two hours per night (*Figure 3*) and the cyclic suppression of heart rates while resting during the day.

The cyclic bradycardia during daytime rest in our study is unprecedented. It is possible that these cycles are linked to REM and pre-REM sleep, but when humans and cats sleep their heart rate slows immediately prior to the elevation of heart rates during REM cycles (*Taylor et al., 1985*; *Verrier et al., 1998*). In both taxa the change in heart rate lasts only seconds and amounts to a total change of 3–5% from the resting heart rate as compared to the minutes-long 30% reduction in *U. bilboatum*. A reversed pattern in heart rate is found in hibernating ground squirrels that show irregular heart rates that speed up for 30–50 s before slowing again to a steady rate (*Milsom et al., 1993*; *Milsom et al., 1999*). The regular occurrence of this cyclic bradycardia suggests that it is a standard

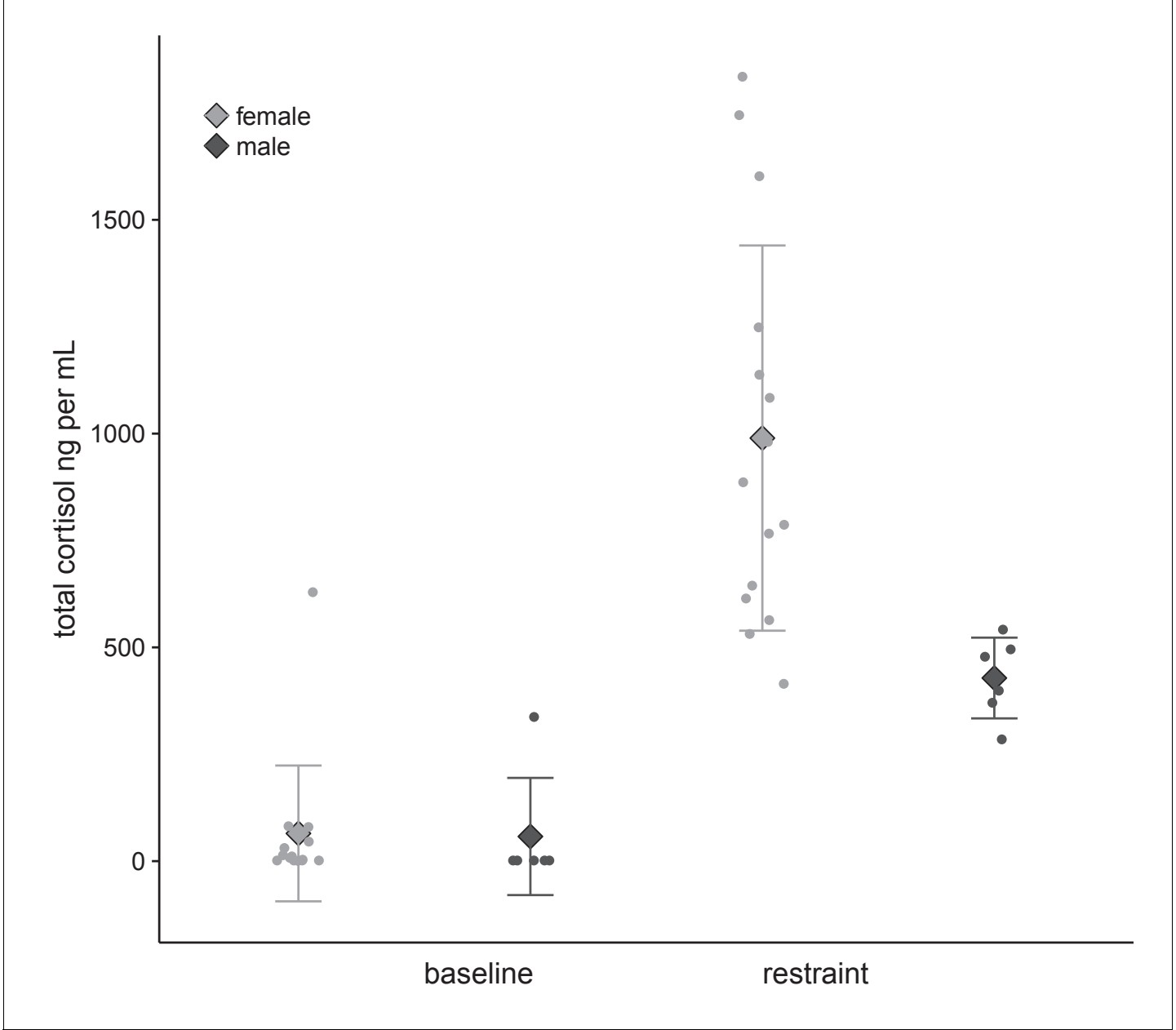

**Figure 5.** Baseline and restrained plasma cortisol values from female (n = 15) and male (n = 6) *U. bilobatum*. There were no differences between sexes in baseline values, but females had higher circulating plasma cortisol values after one hour of restraint.
DOI: https://doi.org/10.7554/eLife.26686.011

and regular aspect of the way that *U. bilobatum* rests and is likely a further extension of the energy conservation of sleep (*Benington and Heller, 1995*; *Kilduff et al., 1993*; *Schmidt, 2014*; *Walker et al., 1979*). Bradycardia is common aspect of the dive response where diving mammals slow their heart rates to conserve oxygen when submerged for long periods (*Noren et al., 2012*). Mammals in torpor are also bradycardic (*Currie et al., 2014*; *Dechmann et al., 2011*; *Elvert and Heldmaier, 2005*; *Heldmaier et al., 2004*), but the cyclic and varying nature of the heart rate depressions we find in *U. bilobatum* are not characteristic of any of these physiological states. Animals enter torpor and hibernation through controlled reductions of heart rate via increased inter-beat interval and skipped beats (*Elvert and Heldmaier, 2005*; *Milsom et al., 1999*). It may be that the slowed heart rates in *U. bilobatum* reflect the initial descent into short and shallow torpor events

with a decrease in heart rate preceding the shift to torpor. Further investigation into the nervous control of bradycardic states in *U. bilobatum* would clarify both how these reductions are executed and any similarity to a sleep-torpor transition (*Milsom et al., 1999*).

Thus far, bats and hummingbirds in torpor and at rest have showed low and constant heart rates without any indication of the cycling we observe (*Currie et al., 2014*; *Dechmann et al., 2011*; *Schaub and Prinzinger, 1999*). Species that are capable of daily torpor generally lower their body temperatures to maximize energetic savings, particularly when exposed to cold temperatures (*Ruf and Geiser, 2015*), and this commonly occurs in tropical and sub-tropical mammals at temperatures below 24°C (*Canale et al., 2012*; *Geiser and Stawski, 2011*). However, a similar torpor response is not possible for the tropical *U. bilobatum*. Instead, they actively defend their body temperatures when exposed to cold, and more than triple metabolic rates to maintain a body temperature of 36°C at ambient temperatures of 10°C vs 30°C (*McNab, 1969*). In our respirometry calibrations, *U. bilobatum* maintained a constant body temperature around 37°C across the measured range of heart rates of 300–800 bpm. In fact, they may alter heart rate dynamics independently of body temperature (*Tøien et al., 2011*). The low heart rates we observed are similar to the minimum resting heart rates of small bats (i.e., 200–400 bpm) in thermoneutral conditions (*Currie et al., 2015*; *Kulzer, 1967*; *Leitner, 1966*; *Leitner and Nelson, 1967*). The thermoneutral zone for *U. bilobatum* is reported to be 29–35°C (*McNab, 1969*; *Rodríguez-Herrera et al., 2016*), which is still slightly higher than the ambient temperature of our field site (mean: 25.87 ± 1.21°C, range: 23.38–28.24°C). It is unclear if the frequency and intensity of the cycling we observed are in response to ecological and energetic interactions, such as lowered foraging success, that decouples resting metabolic rates from overall FMR (*Nilsson, 2002*; *Welcker et al., 2015*). Decoupling resting metabolic rates from total energetic expenditure is hypothesized to be found in animals that live near their energetic ceilings (*Welcker et al., 2015*) with high metabolic rates, like *U. bilobatum*. We found the number of cycles per hour varied both within and among individuals, but we do not yet have enough information on the relationship between total nightly energy expenditure, energy intake, and the lowering of heart rates. However, it is unclear why these bats move between two apparently stable low energy states at rest. Further investigation into the potential relationship between this heart rate change and more commonly perceived torpor states would help understand energetic adaptations in tropical environments.

There are relatively few data on the heart rates of free-flying animals (*Green, 2011*) all of which are larger than *U. bilobatum*. Furthermore, accurately estimating energy consumption during flight under controlled conditions has often been unpractical or impossible due to the conditions needed in wind tunnels and mask respirometry. Our bats' heart rates and metabolic power are surprisingly low and variable when compared to flight tunnel studies. The heart rate derived estimate for cost of flight in *U. bilobatum* (1.36 ± 0.23 W, range 0.98–2.3 W) was slightly lower than mass loss estimates for bats of similar sizes and wing shapes (1.96–2.45 W: (*von Busse et al., 2013*; *Winter and von Helversen, 1998*). However, *U. bilobatum* show a mass specific power of 76 W kg$^{-1}$ with a maximum output of 145 W kg$^{-1}$, which is within the power requirements of bats that are up to 44 times larger (*Carpenter, 1986*; *Thomas, 1975*). While our estimates of energy consumption were not directly calibrated with flying bats, they provide the best potential estimates available based on broad patterns of the relationships among heart rate, stroke volume, and oxygen consumption during exercise (*Bishop and Spivey, 2013*), and must be interpreted with some caution. The large changes in heart rate among activity states remain lower than would be expected based on body size and reinforce the emerging pattern of lower energy consumption by free-flying animals versus those in controlled laboratory conditions (*Bishop et al., 2015*; *Bowlin et al., 2005*; *Ward et al., 2002*). Metabolic power of bat flight may be difficult to predict as a function of body size, but more likely the context in which the animals fly plays a strong role in determining the energy used. Laboratory experiments have been our best window into animals' physiological possibilities, but it is increasingly important to study energetics in relevant ecological settings to understand how these physiological mechanisms evolve.

The ability to rapidly fuel metabolism through ingested food seems to be a common adaptation among hummingbirds and bats with diets of simple carbohydrates and the fastest incorporation rates measured for vertebrates (*Welch et al., 2016*). Nectarivorous hummingbirds (3–5 g) and bats (10 g) fuel 50% of their metabolism within 3–9 min of feeding (*Suarez et al., 2011*; *Voigt and Speakman, 2007*; *Welch et al., 2008*). At three times their body size, *U. bilobatum* shows similar

fractional incorporation rates. In contrast, other fruit-eating bats use incorporation rates of 10–12 min regardless of body size (*Amitai et al., 2010*). These comparative data indicate strong pressure on all flying frugivores, regardless of size, to mobilize ingested food to power flight and this is mediated through paracellular absorption (*Caviedes-Vidal et al., 2008*; *Price et al., 2014*). While initiating flight on stored energy, *U. bilobatum* and other sugar-focused bats rely heavily on ingested carbohydrates to supplement rapidly depleted glycogen at the onset of flight, further taxing the sugar oxidation cascade to push energy to muscle as quickly as possible (*Suarez et al., 2011*; *Welch et al., 2016*). Frugivorous bats deplete the large glycogen stores in their liver within 24 hr (*Pinheiro et al., 2006*). Our fat turnover experiments also showed that half of fat and sugar storage is mobilized within a single day. Specialization on foods rich in simple but rapidly incorporated carbohydrates seems to come with high risks that necessitate additional physiological and behavioral strategies to ensure energetic stability.

*Uroderma bilobatum* further control energetic incorporation and conservation by maintaining exceptionally low baseline cortisol levels that then are elevated to some of the highest recorded naturally induced values for mammals (*Sapolsky et al., 2000*). Basal glucocorticoid values of other bat species are especially high for mammals of their size (100–800 ng ml$^{-1}$; (*Reeder et al., 2004*; *2006*) and show large potential maximal output when challenged with ACTH (*Lewanzik et al., 2012*). However, the difference between baseline and restraint-induced circulating cortisol especially in female *U. bilobatum* is more similar to the extremes found in lemmings (*Lemmus trimucronatus*) that seasonally elevate their baseline corticosterone values by 10–80 times to concentrations of over 4000 ng ml$^{-1}$ (*Romero et al., 2008*) or in flying squirrels (*Glaucomys sp*) that elevate cortisol values 38–40% above already high baseline values (*Desantis et al., 2016*). The low baselines we found may be a consequence of capturing resting or sleeping animals in their day roosts at least 4 hr after sunrise when circulating glucocorticoids were at their lowest (*Sapolsky et al., 2000*). However, this cannot explain peak values 1.5x greater than those observed in other mammals. We suggest that rapid increases in circulating cortisol levels during the acute stress response act in concert to mobilize energy stores, but more importantly, by suppressing glucocorticoid secretion during rest these bats are able to further minimize energetic expenditure and lower their metabolic rates (*Haase et al., 2016*; *Nieuwenhuizen and Rutters, 2008*; *Palme et al., 2005*) and minimize additional fat oxidation (*Brillon et al., 1995*).

Unpredictable fruit availability can have dramatic effects on survival and some bats, including *U. bilobatum* take advantage of their roosts to leverage social information and identify newly available food items (*O'Mara et al., 2014a*; *Ramakers et al., 2016*). Furthermore, the potential for rapid declines in food availability has likely shaped conservative physiological strategies in these bats to minimize energy expenditure while allowing for rapid resource mobilization needed for powered flight. These dynamic energetic strategies likely contribute to the success and diversity of the over 1300 bat species throughout the world (*Simmons, 2005*).

## Materials and methods

All methods were approved by the Autoridad Nacional del Ambiente, Panama (SE/A-88–13; SE/AP-12–14; SE/A-73–14) and by the Institutional Animal Care and Use Committee of the Smithsonian Tropical Research Institute (2012-060-2015; 2014-0701-2017). All data presented are available at the Dryad Digital Repository (doi: 10.5061/dryad.n821p)

### Capture and transmitter attachment

We captured 4 adult *Uroderma bilobatum* (2f/2m, 18.1 ± 1.5 g body mass) from their day roosts in Gamboa, Panamá in December 2014. Bats were fitted with a heart rate transmitter (ca 0.8 g; SP2000 HR Sparrow Systems, Fisher, IL USA) that emitted a continuous long-wave signal modulated by cardiac muscle potentials (*Bowlin et al., 2005*). This added 4.5 ± 0.04% of body mass and is within the range of the additional loading (5%) that should have minimal impact on behavior and physiology of bats and birds (*Aldridge and Brigham, 1988*; *Barron et al., 2010*; *Elliott, 2016*; *O'Mara et al., 2014b*), particularly broad-winged understory foragers like *U. bilobatum*. We trimmed the dorsal fur below the shoulder blades. A topical analgesic was then applied (Xylocaine gel, Astra Zeneca, Wedel Germany) and after disinfecting the electrodes and back of the bats with 70% EtOH, the transmitter's two copper plated gold electrodes were inserted ca. 3 mm through a puncture made with a 23

G sterile needle. The transmitters were mounted on thin, flexible cloth and glued over the electrode insertion points using a silicone-based skin adhesive (Sauer Hautkleber, Manfred Sauer, Germany). The electrodes are flexible and do not appear to disturb the animals, and we expect superficial healing of the small punctures within one hour. While behavioral responses may not directly reflect physiological stress (*Ditmer et al., 2015*), our radio tracking data show typical behavior for this species, and both the large variation and temporal consistent heart rate data we collect do not indicate that bats are either under excessive stress or that habituation was needed to accommodate the added load of the transmitter (*O'Mara et al., 2014b*). After calibration of heart rate versus oxygen consumption (below) animals were tracked for 2–6 days (mean: 3.75 d). We recaptured three of the four bats and removed their transmitters. Bats lost 0.0–0.5 g (0.17 ± 0.29 g) which is within the daily mass fluctuations (1–2 g) observed in this species (O'Mara, unpublished data).

## Calibration of heart rate versus oxygen consumption

We measured rates of oxygen consumption ($\dot{V}O_2$) carbon dioxide production ($\dot{V}CO_2$), heart rate ($f_H$), and body temperature ($T_b$) of these four bats with an open-flow, push-through respirometry system. External air (>75% relative humidity, ~26°C) was dried with Drierite (WH Hammond Driertie Co, Ltd, Xenia, OH, USA) and pumped through a mass flow controller (FB8, Sable Systems International, Las Vegas, NV, USA) into a 1 L respirometry chamber fitted with a thermocouple within a 20 L insulated cooler that was dark and temperature controlled (PELT5, Sable Systems). Flow rate was 600 ml min$^{-1}$, chamber temperature was maintained at 28–29°C, and relative humidity and vapor production were measured with a RH-300 (Sable Systems), and an additional empty chamber served as a reference to the animal chamber. After drying the air leaving the chamber with Drierite we measured $CO_2$ concentration, and after scrubbing the air of $CO_2$ with Ascarite (Thomas Scientific, Swedesboro NJ, USA) we determined $O_2$ concentrations (FOXBOX, Sable Systems). Chamber temperature, $CO_2$, $O_2$, and relative humidity were recorded directly with Expedata via the UI-2 data acquisition interface (Sable Systems). $\dot{V}O_2$ and $\dot{V}CO_2$ were then calculated across five minute intervals (*Lighton, 2008*). Bat $T_b$ was monitored with a temperature sensitive PIT-tag (BioThermo13, Biomark Inc, Boise ID, USA) injected dorsally under the skin and recorded every minute (*Stockmaier et al., 2015*). Bats remained normothermic throughout the experiment with $T_b$ = 36.9 ± 1.6°C (*Figure 1—figure supplement 3*). Heartbeat of bats in the respirometry chamber was recorded as a sound file (see below), and $f_H$ was averaged over the one minute preceding each $T_b$ measurement. This gave five $T_b$ and $f_H$ measures for each measurement of $\dot{V}O_2$ and $\dot{V}CO_2$. After three hours bats were released at their roosts. Respirometry measures were taken between 19:00 – 04:00 hr. Heart rate provided a better fit in a single factor generalized linear mixed effect model (bat identity as a random effect) of energy consumption than body temperature ($R^2_{adjusted}$ = 0.758 vs $R^2_{adjusted}$ = 0.145, respectively), and the inclusion of $T_b$ in a two-factor model did not improve the model's predictive ability. In the best-fit model, energy expenditure was related to $f_H$ as (kJ h$^{-1}$)=0.004 * $f_h$ - 0.3228 (*Figure 1—figure supplement 4*).

## Heart rate telemetry and estimated field energy expenditure

We recorded $f_H$ of the four free-ranging bats during 2–6 days and nights using telemetry receivers (AR8000, AOR Ltd) connected to 3-element Yagi antennae (Sparrow Systems). This was then recorded via mini-dv output to a wave file (44.1–48 kHz) on a digital recorder (Tascam DR-05). Receivers were placed under roosts to record $f_H$ during the full inactive cycle during daylight hours. One to two people then followed the bats at emergence (ca 18:00) for 4–8 hr during the night's activity and continuously recorded estimated activity (flight, inactivity, grooming) via fluctuations in the amplitude of the transmitted signal. Transmitter signal could be detected within 70–100 meters in the forest and up to a kilometer over open space (the Panama Canal). This gave us 18–20 hr of heart rate recordings per individual per day for a total of 350 hr. Daytime mean ambient temperature was 25.87 ± 1.21°C (mean daytime minimum to mean maximum: 23.38–28.24), and mean nightly ambient temperature was 23.74 ± 0.50°C (mean nightly minimum to mean maximum: 22.74–24.78°C). Ambient temperature was recorded by the Autoridad del Canal de Panamá for Gamboa and provided by the Smithsonian Tropical Research Institute's Physical Monitoring Program.

Heart rate from radio transmitters was scored previously by visually measuring the interval needed to encompass 5–10 heart beats at sampling intervals of 0.5–10 min apart (*Barske et al.,*

*2014*; *Bowlin et al., 2005*; *Dechmann et al., 2011*; *Sapir et al., 2010*; *Steiger et al., 2009*). We fully sampled the recorded data using an automated approach in R 3.2 (*Core Team, 2016*) to identify and count all heartbeats (*Figure 1—figure supplement 1*). We used a finite impulse response filter in *seewave* with a window length of 1500–2000 samples to select the carrier frequency of the transmitter. We counted individual heartbeats by applying a timer function in *seewave* that ran over non-overlapping windows of 500 samples. This created a resolution of 88–98 sampling windows per second. We then applied a kernel density filter in *KernSmooth* to further eliminate noise that was outside of the 90% quantile. This approach is conservative in that it may have eliminated some heart rate outliers, but the autocorrelated nature of heart rate allowed us to filter out errors likely induced by static or other interference in the recordings. Automated samples were inspected periodically to validate the filtering method, particularly in periods with high variation.

We then estimated total energy consumption in two ways. First, we used the five minute $\dot{V}CO_2$ production from the respirometry chamber estimate total energy consumption using a conversion of 1 mL $CO_2 \cong 26$ J and matched this to an average of the preceding one minute $T_b$ and $f_H$ measurements. While we attempted to get a range of $f_H$ within the respirometry chamber, we could not attain the high heart rates typical of *U. bilobatum* during flight or the very low $f_H$ we observed during day rest. Furthermore, high $f_H$ of animals due to factors other than exercise, such as our respirometry chamber, may under-estimate energy consumption caused by changes in stroke volume and oxygen extraction efficiency during exercise (*Bishop and Spivey, 2013*). However, we can use the relationship between heart mass ($M_h$, a proxy for stroke volume), and body mass ($M_b$) to model oxygen consumption as function of $f_H$ as $\dot{V}O_2 = 0.0402 M_b^{0.328\pm0.05} M_h^{0.913\pm0.045} f_h^{2.065\pm0.03}$ (*Bishop and Spivey, 2013*). This estimate is based on the exercise response of 24 species of endotherms across 5 orders of magnitude of body size. This model is able to accurately estimate energy consumption during the primary mode of locomotion (*Ward et al., 2002*), which has been the major shortfall of experimental calibration of heart rate against $\dot{V}O_2$ in respirometry conditions where locomotion is restricted. We estimated individual heart mass as 1% of body mass at capture (*Canals et al., 2005*). Because the bulk of *U. bilobatum* diet is carbohydrate we then converted $\dot{V}O_2$ estimates to energy by assuming that 1 ml $O_2 \cong 21.11$ J.

## Metabolic incorporation rates

We used a feeding experiment to measure the change in $\delta^{13}C$ values in exhaled $CO_2$ and estimate the time needed for ingested food to enter metabolic processes and exit as waste $CO_2$. *Uroderma bilobatum* feed on figs with a low enrichment of $^{13}C$, typical for a C3 plant. By feeding an enriched $^{13}C$ source from a CAM plant (agave nectar) we could measure how quickly sugar entered metabolism (*McCue and Welch, 2016*; *Voigt and Speakman, 2007*). Bats were captured from their day roosts and housed individually in mesh-lined cages. At time zero, bats were removed from their cage and immobilized by gently wrapping them in cotton gauze, excluding their heads and feet. They were then placed into a 6 × 6 × 4 cm plastic container with an 18 G needle hermetically attached. After sealing the container, ambient air was washed of $CO_2$ using NaOH and flushed through the plastic container at a flow rate of 700 mL min$^{-1}$. The flushing gas exited the container through the attached needle. The pump was turned off 2 min prior to collection to allow breath to accumulate in the plastic box. To collect accumulated $CO_2$ we pierced the teflon membrane of an exetainer (LabCo Exetainer Buckinghamshire, UK) with the needle tip attached to the plastic container. This vacuumed approximately 4.5 ml of headspace into the vacutainer.

After the initial sample collection (time 0), bats were removed from the container and fed either freshly-collected *Ficus insipida*, or approximately 1.5 ml of a solution of 20% (w/w) agave nectar (Organic Blue Agave Nectar, Wholesome Sweeteners, Sugar Land Texas, USA), 2% (w/w) Nutri-Cal (Vétoquinol Prolab Inc, Princeville, Québec, Canada) and water using a transfer pipette. Breath samples were collected at 0, 10, 20, 30, 40, 60, and 90 min after the initial feeding. Bats fed figs (n = 6) were placed back in their home cage and allowed to feed *ad libitum* after sample collection at 40 mins and 60 mins. Bats fed agave nectar (n = 8) were fed an additional 0.5 ml of the agave nectar solution after sample collection at 30 and 60 min to ensure that the bats' breath was equilibrated isotopically with the new diet. Additional samples at 50 and 70 mins post initial feeding were collected during the agave feeding experiments. Bats fed on figs were returned to their capture site after the last sample collection.

To measure fat turnover following (*Voigt and Speakman, 2007*), bats fed agave nectar were given an additional 1 ml of agave solution after the final sample collection and returned to their home cages. They were maintained on the agave nectar solution supplemented with Nutrical (tomlyn, Fort Worth USA; $\delta^{13}$C Agave + Nutri-Cal: 12.023 ± 0.11 ‰) as their only source of food for the following three nights. Bats were offered agave nectar and water *ad libitum* and were also fed by hand every 3 hr to ensure that they were feeding consistently. Food was removed during the day and bats were fasted for at least 10 hr prior to sample collection. At the beginning of each night were removed from their holding cages to collect a single breath sample as in the previous experiment to measure their baseline $\delta^{13}$C. Following breath collection, bats were fed with the agave nectar solution and returned to their holding cages. Body mass and body condition were monitored throughout the experiment to ensure animal optimum health, and one animal was released after night 2 because of weight loss.

Breath samples were then shipped to the stable isotope laboratory of the Leibniz Institute for Zoo and Wildlife Research where $\delta^{13}$CO$_2$ was analyzed in a blind protocol using a GasBench (Thermo Scientific, Bremen Germany) connected to a stable isotope ratio mass spectrometer (Delta V Advantage Thermo Scientific, Bremen Germany). Samples were analyzed together with a laboratory standard gas that was previously calibrated with the international $^{13}$C reference materials NBS 19 and L-SVEC. Ratios of $^{13}$C and $^{12}$C were expressed relative to the international standard (Vienna-PeeDee Belemnite) using the $\delta$ notation in parts per mill (‰): $\delta^{13}C_{V-PDB} = R_{sample}/R_{standard}-1) \times 10^3$ where $R_{sample}/R_{standard}$ is the ratio of heavy and light carbon isotopes ($^{13}$C/$^{12}$C) in the sample and the standard. Precision was always better than ±0.06‰ (1 SD). To measure the isotopic composition of the agave nectar solution a sample was dried in a drying oven until constant mass and 3 subsamples were then separated, weighed, and loaded in a tin capsule. Samples were analysed together with laboratory standards of known stable carbon isotope ratios using an elemental analyser (Flash elemental analyser, Thermo Scientific, Bremen, Germany) connected in continuous mode via a Conflo III to a stable isotope mass spectrometer (Delta V Advantage; Thermo Scientific, Bremen, Germany). Samples were combusted under chemically pure helium gas in the analyser and resulting gases were then routed to the IRMS for the analysis of stable carbon isotope ratios. The analytical precision was always better than 0.13 per mille (one standard deviation).

We estimated the fractional rate of isotopic incorporation (k) using a one-pool model for each individual bat as $\delta^{13}C_{breath}(t) = \delta^{13}C_{breath}(\infty) + [\delta^{13}C_{breath}(0) - \delta^{13}C_{breath}(\infty)] e^{-kt}$; where $\delta^{13}C_{breath}(t)$ is the isotope composition, $\delta^{13}C_{breath}(\infty)$ is the asymptotic equilibrium isotope composition, and k is the fractional rate of isotope incorporation. The time at which 50% of carbon isotopes are exchanged in the animal's breath is calculated as $t_{50}$=-ln(0.5)/k. The reciprocal of the fractional incorporation rate ($k^{-1}$) estimates the average residence time of a carbon atom in fat reserves. We used a one-compartment model as this typically reflects isotopic incorporation into breath better than models with more complicated dynamics (*Martínez Del Rio and Anderson-Sprecher, 2008*). Non-linear least-squares models based on one-pool dynamics were fit to individual bats.

## Energetic mobilization

We sampled circulating cortisol values from bats (15 F, 6 M) captured from their natural day roosts under the roofs of houses in Gamboa, Panama using a hoop net between 10–12 hr in November 2013 when females were not palpably pregnant or with dependent young. Baseline cortisol samples were collected by puncturing the antebrachial or cephalic vessels with a sterile 23 G needle and collecting ca. 70 uL of blood in heparin-coated hematocrit tubes. Blood samples were collected within 3 min of capture and placed on ice. Bats were placed in a soft cloth bag for 60 min and then a second blood sample of equal volume was collected. After the second blood collection, bats were fed 30% sugar water and released at the capture site. Blood samples were spun in a centrifuge for seven minutes at 7000 g, the plasma removed and snap frozen in liquid nitrogen before storage at −30°C. Samples were then mailed on dry ice to Trent University where total plasma cortisol was measured in duplicate using a commercially available radioimmunoassay (MP Biomedicals ImmuChem Coated Tube Cortisol $^{125}$I RIA Kit; MP Biomedicals, LLC, Diagnostic Division, Orangeburg, NY, USA). This kit was validated for parallelism with plasma from *U. bilobatum*. Tests for differences between slopes on log-transformed data showed that the serially diluted plasma curve was parallel to the assay standard curve ($F_{1,9}$ = 0.21, p=0.66). The intra-assay coefficient of variation (CV) was 2.4% and all samples were run in a single assay. Seven generalized linear mixed effects models in *lme4* were used to

evaluate circulating cortisol concentrations (ng ml$^{-1}$) for effects of sex and time point sampled (baseline or restraint) using animal identity as a random effect.

## Acknowledgements

We would like to thank Rachel Page, the Gamboa Bat Lab, the Smithsonian Tropical Research Institute, and the Autoridad del Canal de Panamá for facilitating this work. Tom Faughnan, Bart Kranstauber, Inge Müller, Nele Herdina, Sebastian Stockmaier, and Sebastian Rikker helped with data collection. We would also like to thank the homeowners in Gamboa for allowing us access to their homes, particularly Hubert Herz, Daisy Dent, . Steve Paton (STRI) provided the daily ambient temperature data for Gamboa. We are grateful to Karin Grassow who analyzed the breath samples for stable isotope ratios, and to Sharon Swartz who provided valuable insight on a previous draft. This work was funded, in part, by the National Geographic Society (GEFNE124-14), The Max Planck Society, the Marie Skłodowska-Curie Actions, and the University of Konstanz.

## Additional information

### Funding

| Funder | Grant reference number | Author |
| --- | --- | --- |
| National Geographic Society | GEFNE124-14 | M Teague O'Mara<br>Martin Wikelski<br>Dina KN Dechmann |
| Max Planck Institute for Ornithology | | M Teague O'Mara<br>Martin Wikelski<br>Dina KN Dechmann |
| University of Konstanz | | M Teague O'Mara<br>Martin Wikelski<br>Dina KN Dechmann |
| Marie Sklodowska-Curie Actions | | M Teague O'Mara |
| Natural Sciences and Engineering Research Council of Canada | RGPIN-04158-2014 | Gary Burness<br>Lanna M Desantis |

The funders had no role in study design, data collection and interpretation, or the decision to submit the work for publication.

### Author contributions

M Teague O'Mara, Conceptualization, Resources, Data curation, Software, Formal analysis, Supervision, Funding acquisition, Validation, Investigation, Visualization, Methodology, Writing—original draft, Project administration, Writing—review and editing; Martin Wikelski, Conceptualization, Resources, Supervision, Funding acquisition, Investigation, Methodology, Writing—original draft, Writing—review and editing; Christian C Voigt, Conceptualization, Resources, Formal analysis, Funding acquisition, Validation, Investigation, Methodology, Writing—review and editing; Andries Ter Maat, Resources, Software, Formal analysis, Investigation, Methodology, Writing—review and editing; Henry S Pollock, Resources, Formal analysis, Methodology, Writing—review and editing; Gary Burness, Conceptualization, Resources, Supervision, Funding acquisition, Investigation, Methodology, Project administration, Writing—review and editing; Lanna M Desantis, Formal analysis, Investigation, Methodology, Writing—original draft; Dina KN Dechmann, Conceptualization, Supervision, Funding acquisition, Investigation, Methodology, Writing—original draft, Project administration, Writing—review and editing

### Author ORCIDs

M Teague O'Mara http://orcid.org/0000-0002-6951-1648
Gary Burness http://orcid.org/0000-0002-1695-7179

Dina KN Dechmann [ID] http://orcid.org/0000-0003-0043-8267

### Ethics

Animal experimentation: All methods were approved by the Autoridad Nacional del Ambiente, Panama (SE/A-88-13; SE/AP-12-14; SE/A-73-14) and by the Institutional Animal Care and Use Committee of the Smithsonian Tropical Research Institute (2012-060-2015; 2014-0701-2017).

### Decision letter and Author response

Decision letter https://doi.org/10.7554/eLife.26686.017
Author response https://doi.org/10.7554/eLife.26686.018

## Additional files

### Supplementary files

• Supplementary file 1. Mean ± sd per bat for heart rates (frequency in beats per minutes, $f_H$) and energy consumption (kilojoules per hour, kJ h$^{-1}$ and Watts, W) used during flight, roosting at night, and resting during the day. N gives the number of observations of heart rate observations used to calculate values.
DOI: https://doi.org/10.7554/eLife.26686.012

• Supplementary file 2. Individual fitted equations of a 1 pool exponential model for incorporation of ingested $\delta^{13}C$ into breath carbon dioxide. 1 pool model: $\delta^{13}C$ breath$(t)$ = $\delta^{13}C$ breath$(\infty)$ + $[\delta^{13}C$breath$(0) - \delta^{13}C$ breath$(\infty)]$ e$^{-t/k}$
DOI: https://doi.org/10.7554/eLife.26686.013

• Transparent reporting form
DOI: https://doi.org/10.7554/eLife.26686.014

### Major datasets

The following dataset was generated:

| Author(s) | Year | Dataset title | Dataset URL | Database, license, and accessibility information |
|---|---|---|---|---|
| M Teague O'Mara, Martin Wikelski, Christian C Voigt, Henry S Pollock, Lanna M Desantis, Gary Burness, Andries Ter Maat, Dina KN Dechmann | 2017 | Cyclic bouts of extreme bradycardia counteract the explosive metabolism of frugivorous bats | http://dx.doi.org/10.5061/dryad.n821p | Available at Dryad Digital Repository under a CC0 Public Domain Dedication |

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
