## [Decision Letter]

Thank you for submitting your article "Cyclic bouts of extreme bradycardia counteract the explosive metabolism of frugivorous bats" for consideration by *eLife*. Your article has been reviewed by three peer reviewers, one of whom is a member of our Board of Reviewing Editors and the evaluation has been overseen by the Reviewing Editor and Ian Baldwin as the Senior Editor. The following individual involved in review of your submission has agreed to reveal his identity: Fritz Geiser (Reviewer #2).

The reviewers have discussed the reviews with one another and the Reviewing Editor has drafted this decision to help you prepare a revised submission. Whereas we realize the number of essential revisions is rather extensive in this particular case, we believe that rewriting this manuscript such that it is more rigorous is entirely feasible based on the excellent style of the submitted manuscript. In general, we are mostly concerned with overstatements and a lack of essential information needed to place the findings in context; we do not question your key field observations.

Please comply to your best ability to the comments and provide point-by-point responses indicating at the start of every response if you comply, explain, or respectfully disagree.

Summary:

The manuscript by O'Mara et al., reports the discovery of a new cyclic bradycardic state in tropical fruit bats that saves energy by reducing the heart rate to extremely low values for a series of short moments during day rest. This is a unique and exciting find that advances our understanding of the mammalian cardiovascular system and metabolism. The technical accomplishment of measuring heart rate in freely flying individuals over several days in their natural habitat is impressive. Overall the use of heart rate as a measure of energy expenditure in small free-ranging mammals is commendable, the study was well constructed and the manuscript well written. The findings will be of interest to physiologists, ecologists, comparative biomechanists and organismal biologists in general.

Essential revisions:

Overall, we believe that the manuscript needs some careful reorganization to ensure scientific rigor: That the discussion and conclusions are placed within the context of the studies limitations.

Please discuss how estimates of energy expenditure and energy savings are extrapolations from a small sample of HR and MR under limited conditions and a single activity state. Further, the initial calibrations of HR against MR seem limited in that they were only conducted over a short time frame (3hrs) and at a single high ambient temperature very close to the thermoneutral zone. This gives a likelihood that animals with high HR (around 800bpm may have been exhibiting stress associated with the respirometry procedure, which can impact the derived regression equation as stress response alters blood pressure and cardiac output and is not indicative of true resting conditions.

While we agree that these short bouts of bradycardia are unlikely to be representative of torpor. We note that even small reductions in Tb can be reflected in reductions in HR and energy savings. Especially, because at rest many tropical and subtropical bats can reduce their body temperature by up to 6 degrees.

We appreciate the authors may make inferences of the costs of flight based on their resting calibration, however, these extrapolations may be inaccurate and this should be discussed appropriately. Alternatively, the paragraph in the discussion about flight costs (in the Discussion section) could simply be removed altogether without detracting from the findings improving the overall rigor of the manuscript.

To assist the reader please clarify how the regression calculations were conducted. How long were the HR values averaged with VO2? What time of day were the animals placed in the respirometry chamber? And were issues of autocorrelation and repeated measures addressed in the calculation of the regression equation?

Please further clarify the method by which HR was analyzed so the reader does not need to find and read the Cochran and Wikelski, 2005 reference but can simply fully rely on the Materials and methods section of the present manuscript to appreciate both its strengths and limitations.

Please clarify how many male and female bats were sampled for circulating cortisol, reading the manuscript we were confused by the different statements.

Title: An explosive metabolism would interfere with survival of the organisms. Another adjective may be more appropriate.

Bat stress related comments:

Based on the Introduction we wondered how well the bats function with the backpack after the surgical procedure. This also holds for the discussion in the Introduction and Discussion section "truly evolve" (too strong wording). Both the previous wind tunnel and invasive field experiments performed here are stressful to the animals unless habituation and positive reinforcement training have been implemented with all stressors removed. Only non-invasive non-stress experiments may possibly give this insight.

Reading the Materials and methods the 23ga needle is rather large, the metabolism of the bats is high, how quickly are they expected to heal up compared to the duration of the experiment? Please discuss this in the Materials and methods.

The Materials and methods states the backpack weighs 0.8 gram, which seems a high percentage of body weight. Heavy backpacks are a continues stressor of an animal with a high metabolic rate close to starvation. Please give the percentage backpack weight versus body weight and discuss it in the context of bat and bird experiments in which the effect of backpacks on stress, metabolics and general energetics have been studied. Please integrate this in the discussion if the backpack weighs more than 5% (limit for bird studies without recapture) bodyweight, otherwise integration in the Materials and methods section is sufficient.

Figure 2; there appears to be a data gap beyond 22:00 hours, please explain and discuss. Panel B also appears to overlap with the data of panel A causing another gap, or there is another gap. Please plot the entire data of panel A without gaps from 0-24hrs if possible, otherwise indicate gaps using a gray area / bar with the explanation of why gaps exist.

Materials and Methods: What is the body mass and sample size of the bats? Sample sizes also need to be integrated into the Results section. Also include the average mass and std when discussing backpack mass in the Materials and methods and discuss the size of the backpack with respect to the size measure of the bat.

Table 1: include the mass and a measure of size of the individuals.

Abstract: Clearly much more than the cardiovascular system is involved here, for example digestion among other functions will be crucial in all this.

Introduction: Statement on homeostasis. There are other approaches than simply maintaining homeostasis in animal physiology.

Introduction:. …15-16 times 'of' minimum metabolic rates. Is this in comparison to BMR? If so, state. BTW some small bats at rest and without flight can do 300-400 times of the minimum metabolic rate during torpor.

Introduction: It is more complex than just lowering heart rate and body temperature and bats can reduce metabolic rate during torpor by up to 99% even when compared to BMR and more when compared to RMR at low ambient temperatures. Rephrase.

Introduction: There are published data on reduction of metabolism by 50% within the TNZ. Moreover, tropical bats are not always in thermo-neutrality, see the above reference. The sentence also does not make sense as written because they should be able to lower body temperature outside of thermo-neutral conditions. And reductions in heart rates will not only reduce metabolism. Rephrase.

Introduction: quadratic?

Introduction: Here it appears that all of the energy comes from food, further down stores of glycogen are needed. See also your results. Rephrase.

Introduction: By manipulating circulating levels of what?

Introduction: Define 'central-place forager'

Introduction: The reference in the Indroduction lists a number of phyllostomids using torpor (some described by McNab from lab studies as homeothermic) and also other nectarivorous/frugivorous bat.

Introduction: How would not using torpor result in a high metabolic scope -- should it not be the opposite?

Introduction: Further up it is argued that lab work is no good, now lab work is used. An explanation as to why this needs to be done in the laboratory would help.

Results: Large range in heart rates. To put this into perspective, six–fold is pretty good, but humans can do about four–fold and hibernating bats about eighty–fold.

Results: Were these bats captured or in captivity? If so, where and how?

Discussion: Entire range? Did they not fly over the canal?

Discussion: Replacement of half the fat reserves. The results on this need to be made clearer.

Discussion: 10% is not a lot and perhaps insignificant with regard to FMR.

Discussion: The short foraging times are interesting and are known from other small mammals in the field.

Discussion: Predictions of FMR from body mass?

Discussion: See comments above regarding McNab and reference in the Introduction. Differences between field and laboratory data are not only observed for flight, but also torpor expression and there is even a review on that. Rephrase this section.

Discussion: High sustained metabolic rates. Further up in the Discussion section it is stated their FMR is low.

Discussion: How will cyclic bradycardia maintain a semi-vigilant (and why semi?) state?

Discussion: Fuel metabolism from what?

Discussion: Is 50% of fat a majority?

Materials and methods: The statement is insufficient. In bears it was observed that behavioral responses to stressors can be minimal even if physiological responses are significant:http://www.cell.com/current-biology/abstract/S0960-9822(15)00827-1 Please discuss these issues briefly and fairly.

Materials and methods: Please include detailed information on the following: were animals retrieved during the experiment or after the experiment? Was the heart rate transmitter removed during or after the experiment, was the animal sacrificed or released after the experiment? If it was released in which state was the animal released?

Materials and methods: How was body temperature tracked? Provide details.

Materials and methods: What were the ambient temperatures during field work? If you can, provide daily minima, maxima and averages.

Materials and methods: Address assumptions and define abbreviations.

Materials and methods: Were the Figures and agave nectar analyzed for isotopes?

Materials and methods: This needs more explanation.

Materials and methods: What was in the vacutainers?

Materials and methods: This was queried above, were these in the wild? If so, please describe how bats were found, captured etc.

[Editors' note: further revisions were requested prior to acceptance, as described below.]

Thank you for resubmitting your work entitled "Cyclic bouts of extreme bradycardia counteract the high metabolism of frugivorous bats" for further consideration at *eLife*. Your revised article has been favorably evaluated by Ian Baldwin (Senior editor), a Reviewing editor, and two reviewers.

The manuscript has been improved but there are some remaining issues that need to be addressed before acceptance, as outlined below:

1) Throughout the results and discussion energy expenditure in free ranging bats should be clarified as estimated, calculated or "HR derived" or equivalent. (E.g. Discussion section and other lines.)

2) The bradycardia experienced by these animals at rest seem to fall within the range expected for basal heart rate and values found for other species of bats of a similar size or smaller (see Kulzer, 1967 and Currie et al., 2015), therefore related statements should be clarified accordingly.

---

## [Author Response]

Essential revisions:Overall, we believe that the manuscript needs some careful reorganization to ensure scientific rigor: That the discussion and conclusions are placed within the context of the studies limitations.Please discuss how estimates of energy expenditure and energy savings are extrapolations from a small sample of HR and MR under limited conditions and a single activity state. Further, the initial calibrations of HR against MR seem limited in that they were only conducted over a short time frame (3hrs) and at a single high ambient temperature very close to the thermoneutral zone. This gives a likelihood that animals with high HR (around 800bpm may have been exhibiting stress associated with the respirometry procedure, which can impact the derived regression equation as stress response alters blood pressure and cardiac output and is not indicative of true resting conditions.

This is an excellent point, and one we have attempted to circumvent using the allometric relationships based on heart and body mass developed by Bishop & Spivey (2013). Under stress conditions, it is unclear what the relationship between heart rate and oxygen consumption truly is. First, as a semantic distinction it is unclear why the reviewers assume that only high heart rates are indicative of stress. In our experience with frugivorous phyllostomids, such as *Uroderma bilobatum*, the most dangerous and stressful situations are when these animals enter low energy states, reduce T_b_ and f_H_. While rare, in these stress-induced states, animals are unlikely to recover. Furthermore, high stress can induce torpor conditions in many, including tropical, bat species (e.g., Molossidae). The physiological adjustments animals make to increase blood pressure, mobilize glucose reserves, and hyperventilate, to name only a few, all could yield contradictory energetic results. Furthermore, increased heart rate in these conditions may not have the full cardiac expansion and contraction of exercising heart rates.

One point of confusion that we would like to highlight is how we extrapolated field metabolic rates from the heart rate data. There has been quite a bit of discussion in the literature about the applicability and limitations of respirometry-based calibration of f_H_ since most animals in these situations are not capable of exercising. Therefore, the calibrations of f_H_ that we, and many others have used, fail to include exercise values of animals engaging in their typical modes of locomotion. The allometric equations developed by Bishop and Spivey, (2013) are a breakthrough in overcoming this limitation. Instead of the resting-state calibrations we conducted, we instead apply Bishop and Spivey's relationships using body mass and heart mass of our bats. This was noted in the original submission in subsection “Heart rate telemetry and estimated field energy expenditure”, but it was not made clear that this is the fundamental relationship that defined the energetic expenditure of our field study. This has been applied to studies on bar-headed geese (Bishop et al., 2015, Science doi: 10.1126/science.1258732; Hawkes et al., 2014, PloS one doi: 10.1371/journal.pone.0094015), great frigate birds (Wiemerskirch et al., 2016; Science doi: 10.1126/science.aaf4374), and golden-collared manakins (Barske et al., 2014, Proc R Soc B doi: 10.1098/rspb.2013.2482) We have clarified this in the Materials and methods, the Introduction, and in the Results.

Resting calibrations of animals in respirometry have been widely applied using the heart rate methods, and likely over-estimate energy consumption at the low ends of the calibration values, and more importantly dramatically under-estimate energy consumption at high heart rates. Bishop and Spivey, (2013) highlights the work by Dechmann and colleagues (2011) and the implausibly low field metabolic rates due to underestimates of flying energy consumptions via exercise restricted heart rate / metabolic rate calibrations. Outside of Bishop and Spivey’s particularly thorough work, the quadratic relationship driving this is nicely shown by Ward et al., (2002) and the differences in metabolic rates of geese under their primary and non-primary modes of exercise locomotion. There are few data from flying vertebrates that have captured this so completely.

While we agree that these short bouts of bradycardia are unlikely to be representative of torpor. We note that even small reductions in Tb can be reflected in reductions in HR and energy savings. Especially, because at rest many tropical and subtropical bats can reduce their body temperature by up to 6 degrees.

The use of torpor and heterothermy by tropical and subtropical bats when ambient temperature drops below 24°C is an important detail and we have noted this in the Discussion section. In other work, we have even found that a free-tailed bat species in Panama is capable of reducing energetic expenditure to values lower than the minimal torpor metabolic rates of many sub-tropical and temperate zone mammals, but at body temperatures greater than 32°C, largely by lowering heart rate independently of lowered body temperatures (O’Mara et al., in review, Proc R Soc B). We also note in this paragraph that at least in previous work with captive animals by McNab, *U. bilobatum* actively defends body temperatures and will not undergo heterothermy. It would have been ideal if we were able to also simultaneously measure body temperature of free-ranging bats, but their roosts would not accommodate the PIT tag antenna.

We appreciate the authors may make inferences of the costs of flight based on their resting calibration, however, these extrapolations may be inaccurate and this should be discussed appropriately. Alternatively, the paragraph in the discussion about flight costs (in the Discussion) could simply be removed altogether without detracting from the findings improving the overall rigor of the manuscript.

Flight is obviously the most energetically demanding aspect of a bat’s daily life, and with so few data from free-flying animals we feel that discussing this is worthwhile. We have added a qualifying statement of the difficulty of accurately measuring in-flight metabolic rates in subsection “Heart rate telemetry and estimated field energy expenditure”.

To assist the reader please clarify how the regression calculations were conducted. How long were the HR values averaged with VO2? What time of day were the animals placed in the respirometry chamber? And were issues of autocorrelation and repeated measures addressed in the calculation of the regression equation?We have clarified this in the Materials and methods to show that heart rates were averaged over the one minute preceding the VO2 measure, as well as the time of day when the experiments were conducted (subsection “Metabolic incorporation rates.”). To account for potential confounds of repeated measures we included individual identity as a random effect. We did not attempt to correct for autocorrelation in the heart rate data, but treated the five-minute samples as independent units.Please further clarify the method by which HR was analyzed so the reader does not need to find and read the Cochran and Wikelski, 2005 reference but can simply fully rely on the Materials and methods section of the present manuscript to appreciate both its strengths and limitations.

We are unclear to what this comment is referring since the 2005 Cochran & Wikelski chapter focuses on the spatial tracking of thrushes, not physiological sampling. We have added the sampling windows and intervals for how heart rate was previously visually counted in subsection “Metabolic incorporation rates.” so that it is clear how our current method differs.

Please clarify how many male and female bats were sampled for circulating cortisol, reading the manuscript we were confused by the different statements.

This is noted in subsection “Glucocorticoids & energy mobilization” and in Figure 4.

Title: An explosive metabolism would interfere with survival of the organisms. Another adjective may be more appropriate.

We have adjusted the title accordingly to read ‘high metabolism’.

Bat stress related comments:Based on the Introduction we wondered how well the bats function with the backpack after the surgical procedure. This also holds for the discussion in the Introduction and Discussion sections "truly evolve" (too strong wording). Both the previous wind tunnel and invasive field experiments performed here are stressful to the animals unless habituation and positive reinforcement training have been implemented with all stressors removed. Only non-invasive non-stress experiments may possibly give this insight.

The impact of tag placement on an animal’s behavior is always of concern, and it is unlikely that any tag attachment, no matter how small or streamlined will have no impact on an animal. The best we can do is to minimize this impact on the limited number of animals that we study. The 0.8 g tag we used represents 4-5% of body mass (4.5 ± 0.04%). We have included this summary of body mass and tag mass as a percent of body mass in the Discussion section. We have also revised the wording for the statements. For all studies conducted in wind tunnels extensive habituation and positive reinforcement are needed to get animals to fly in these environments. Animals must be trained to use these devices. We therefore need to acknowledge that the results of these wind tunnel experiments represent normal unstressed states for animals under those conditions. As we note throughout the manuscript, however, the more we learn from free-ranging animals, even those wearing any biologging device, the more we understand that there is great behavioral, mechanical, and physiological flexibility that we have underestimated in these study species.

Reading the Materials and methods the 23ga needle is rather large, the metabolism of the bats is high, how quickly are they expected to heal up compared to the duration of the experiment? Please discuss this in the Materials and methods.

Healing time is difficult to estimate, but we also don't believe that the 23ga needle is large (⌀ = 0.6414 mm). The needle punctures are difficult to find for lead insertion and we expect superficial healing within an hour. In comparison, human insulin needles are typically 26–31 ga (⌀ = 0.4636–0.2604 mm), and most intramuscular and subcutaneous injections for human infants are done with 23ga needles. Furthermore, cannulae used to inject PIT tags that are commonly used on small mice and bats including *Uroderma* in our study populations are 12ga ((⌀ = 2.769) and no negative impact has been observed. We have added additional statement regarding potential healing time in subsection “Calibration of heart rate versus oxygen consumption.”

The Materials and methods states the backpack weighs 0.8 gram, which seems a high percentage of body weight. Heavy backpacks are a continues stressor of an animal with a high metabolic rate close to starvation. Please give the percentage backpack weight versus body weight and discuss it in the context of bat and bird experiments in which the effect of backpacks on stress, metabolics and general energetics have been studied. Please integrate this in the discussion if the backpack weighs more than 5% (limit for bird studies without recapture) bodyweight, otherwise integration in the Materials and methods section is sufficient.

This additional mass represents 4.5 ± 0.04% of body mass. This, and the general limits of additional loading on flying animals, has been added in the Material and methods.

Figure 2; there appears to be a data gap beyond 22:00 hours, please explain and discuss. Panel B also appears to overlap with the data of panel A causing another gap, or there is another gap. Please plot the entire data of panel A without gaps from 0-24hrs if possible, otherwise indicate gaps using a gray area / bar with the explanation of why gaps exist.

There are indeed gaps in the data. We have amended Figure 2 to note these gaps and explain in the figure that this is when bats were out of range of radio telemetry (i.e., they flew faster through the forest than we could run) or when they left the roost when recording had been automated.

Materials and Methods: What is the body mass and sample size of the bats? Sample sizes also need to be integrated into the Results section. Also include the average mass and std when discussing backpack mass in the Materials and methods and discuss the size of the backpack with respect to the size measure of the bat.

Noted in subsection “Heart rate telemetry and estimated field energy expenditure”. The sample sizes, both in the number of individuals tracked (or sampled) and the duration of tracking, were integrated in each section in subsections “Activity patterns”; “Field metabolic rates and cyclic bradycardia“, “Metabolic incorporation rates of resting bats“; and Figure 4.

Table 1: include the mass and a measure of size of the individuals.

Added.

Abstract: Clearly much more than the cardiovascular system is involved here, for example digestion among other functions will be crucial in all this.

This has been deleted to simplify.

Introduction: Statement on homeostasis. There are other approaches than simply maintaining homeostasis in animal physiology.

Revised to ‘maintain physiological integrity’.

Introduction:. …15-16 times 'of' minimum metabolic rates. Is this in comparison to BMR? If so, state. BTW some small bats at rest and without flight can do 300-400 times of the minimum metabolic rate during torpor.

Clarified to resting metabolic rate.

Introduction: It is more complex than just lowering heart rate and body temperature and bats can reduce metabolic rate during torpor by up to 99% even when compared to BMR and more when compared to RMR at low ambient temperatures. Rephrase.

Rephrased

Introduction: There are published data on reduction of metabolism by 50% within the TNZ. Moreover, tropical bats are not always in thermo-neutrality, see the above reference. The sentence also does not make sense as written because they should be able to lower body temperature outside of thermo-neutral conditions. And reductions in heart rates will not only reduce metabolism. Rephrase.

Rephrased.

Introduction: quadratic?

Corrected.

Introduction: Here it appears that all of the energy comes from food, further down stores of glycogen are needed. See also your results. Rephrase.

Rephrased.

Introduction: By manipulating circulating levels of what?

Clarified.

Introduction: Define 'central-place forager'

Clarified.

Introduction: The reference in the Indroduction lists a number of phyllostomids using torpor (some described by McNab from lab studies as homeothermic) and also other nectarivorous/frugivorous bat.

This is true; however; McNab has previously shown that this species does not use torpor.

Introduction: How would not using torpor result in a high metabolic scope -- should it not be the opposite?

The breadth of metabolic scope is not a focus for our MS, so we have removed it from this statement.

Introduction: Further up it is argued that lab work is no good, now lab work is used. An explanation as to why this needs to be done in the laboratory would help.

We did not intend to say that lab work is no good – only that what has been seen is that there are substantial differences between the responses of the same species under laboratory conditions and when they are placed into more complicated ecological circumstances. In addition to the work we highlight, this has also been shown by Professor Geiser’s group, and others, regarding torpor and homeothermy expression (e.g, Geiser et al. 2007 doi:10.1007/s00360-007-0147-6; Geiser et al. 2000 doi:10.1007/978-3-662-04162-8_10; and Audet & Thomas 1997 doi: 10.1007/s003600050058). We have added a sentence in the Introduction to hopefully minimize these concerns and reaction about the importance of controlled experiments.

Results: Large range in heart rates. To put this into perspective, six–fold is pretty good, but humans can do about four–fold and hibernating bats about eighty–fold.

Thank you for the additional context.

Results: Were these bats captured or in captivity? If so, where and how?

Clarified.

Discussion: Entire range? Did they not fly over the canal?

This sentence highlights the range of activities, not necessarily spatial range. We have revised accordingly. We did not track the bats across the canal, but were able to maintain radio contact with them due to the large open space and that the bats apparently did not venture far into the forest on the south side of the canal.

Discussion: Replacement of half the fat reserves. The results on this need to be made clearer.

We believe that the results for this are clear in subsection “Glucocorticoids & energy mobilization” and Figure 4, but we would appreciate further guidance as to what should be clarified.

Discussion: 10% is not a lot and perhaps insignificant with regard to FMR.

We respectfully disagree, particularly in light of the rapid incorporation rates and fat turnover that we also document.

Discussion: The short foraging times are interesting and are known from other small mammals in the field.

We agree, and this seems to be the norm for other frugivorous bats as well.

Discussion: predictions of FMR from body mass?

Clarified.

Discussion: see comments above regarding McNab and reference in the Introduction. Differences between field and laboratory data are not only observed for flight, but also torpor expression and there is even a review on that. Rephrase this section.

We are unclear about the specifics of this comment as the reference in the Discussion section focuses on past work by McNab which shows that U. bilobatum defends Tb across a wide range of temperatures and does not enter classic torpor. We have added reference in the previous sentence that is more focused on lowering Tb in torpor to also include tropical and sub-tropical species. We would like to note as well that torpor appears to be more likely to be document in laboratory conditions than in free-ranging animals.

Discussion: High sustained metabolic rates. Further up in the Discussion section it is stated their FMR is low.

It is high for a mammal of their size, but what is expected for a frugivorous bat. We have rephrased this.

Discussion: How will cyclic bradycardia maintain a semi-vigilant (and why semi?) state?

We have removed this statement.

Discussion: Fuel metabolism from what?

Clarified.

Discussion: Is 50% of fat a majority?

Rephrased.

Materials and methods: The statement is insufficient. In bears it was observed that behavioral responses to stressors can be minimal even if physiological responses are significant:http://www.cell.com/current-biology/abstract/S0960-9822(15)00827-1 Please discuss these issues briefly and fairly.

This is an excellent point and the citation is spot on, particularly since it includes heart rate data. We have added further discussion in the Materials and methods section.

Materials and methods: Please include detailed information on the following: were animals retrieved during the experiment or after the experiment? Was the heart rate transmitter removed during or after the experiment, was the animal sacrificed or released after the experiment? If it was released in which state was the animal released?

3 of the 4 bats were recaptured and the transmitter was removed. Bats lost 0.0–0.5g (0.17 ± 0.29 g) which is within the daily mass fluctuations of 1–2g observed in this species (487 bats 92 of which have been captured more than once; O’Mara, unpublished data).

Materials and methods: How was body temperature tracked? Provide details.

This was described in subsection “Calibration of heart rate versus oxygen consumption” of the original submission. There may be confusion with the word "tracked" in that sentence. We did not measure Tb in free-ranging animals, only in respirometry. This has been clarified.

Materials and methods: What were the ambient temperatures during field work? If you can, provide daily minima, maxima and averages.

Added in subsection “Energetic Mobilization”.

Materials and methods: address assumptions and define abbreviations.

Added.

Materials and methods: Were the Figures and agave nectar analyzed for isotopes?

We measured the value of the agave nectar solution but not the Figures.

Materials and methods: This needs more explanation.

We have expanded these methods and hopefully they are now clear.

Materials and methods: What was in the vacutainers?

Rephrased.

Materials and methods: This was queried above, were these in the wild? If so, please describe how bats were found, captured etc.

We have clarified these methods: These were all wild bats in their natural roost.

[Editors' note: further revisions were requested prior to acceptance, as described below.]

The manuscript has been improved but there are some remaining issues that need to be addressed before acceptance, as outlined below:1) Throughout the results and discussion energy expenditure in free ranging bats should be clarified as estimated, calculated or "HR derived" or equivalent. (E.g. Discussion section and other lines.).

We have corrected this accordingly by adding ‘heart rate derived’ where appropriate (e.g., Discussion section).

2) The bradycardia experienced by these animals at rest seem to fall within the range expected for basal heart rate and values found for other species of bats of a similar size or smaller (see Kulzer 1967 and Currie et al., 2015), therefore related statements should be clarified accordingly.

This is true and is it interesting how consistent basal heart rates are in bats. We have added detail related to this in the Discussion section that gives the range basal heart rate of small bat species and also notes the thermoneutral zone for *U. bilobatum* relative to the ambient temperatures of our field site for readers to consider.